# Landslide Displacement Prediction Based on Multivariate LSTM Model

**DOI:** 10.3390/ijerph20021167

**Published:** 2023-01-09

**Authors:** Gonghao Duan, Yangwei Su, Jie Fu

**Affiliations:** 1School of Computer Science and Engineering, Wuhan Institute of Technology, Wuhan 430205, China; 2Engineering Research Center of Intelligent Technology for Geo-Exploration, Ministry of Education, Wuhan 430074, China; 3Hubei Key Laboratory of Advanced Control and Intelligent Automation for Complex Systems, Wuhan 430074, China; 4Hubei Provincial Key Laboratory of Intelligent Robot, Wuhan 430205, China; 5Center For Hydrogeology and Environmental Geology Survey, China Geological Survey, Baoding 071051, China

**Keywords:** landslide, displacement prediction, cubic spline interpolation, multivariate LSTM, Three Gorges Reservoir area

## Abstract

There are many frequent landslide areas in China, which badly affect local people. Since the 1980s, there have been more than 200 landslides in China with a death toll of 30 or more people at a time, economic losses of more than CNY 10 million or significant social impact. Therefore, the study of landslide displacement prediction is very important. The traditional ARIMA and LSTM models are commonly used for forecasting time series data. In our study, a multivariable LSTM landslide displacement prediction model is proposed based on the traditional LSTM model, which integrates rainfall and reservoir water level data. Taking the Baijiabao landslide in the Three Gorges Reservoir area as an example, the data of displacement, rainfall and reservoir water level of monitoring point ZG323 from November 2006 to December 2012 were selected for this study. Our results show that the displacement prediction results of the multivariable LSTM model are more accurate than those of the ARIMA and the univariate LSTM models, and the mean square, root mean square and mean absolute errors are the smallest, which are 0.64223, 0.8014 and 0.50453 mm, respectively. Therefore, the multivariable LSTM model method has higher accuracy and better application prospects in the displacement prediction of the Baijiabao landslide, which can provide a certain reference for the displacement prediction of the same type of landslide.

## 1. Introduction

In recent years, with the frequent occurrence of extreme weather, the risk of landslides has increased. According to the statistics of the Ministry of Natural Resources, there were 579 landslides in China in the first half of this year that resulted in 17 deaths, three missing people and direct economic losses of CNY 240.8 million. Therefore, there is an urgent need to predict landslide displacement data for time series to reduce losses [1].

Landslide displacement prediction is currently a frontier issue in the field of international landslide research [2]. Since the 1980s, mathematical models have been widely used to fit the displacement-time curve of monitoring points for analysis so as to predict the dynamic trend of landslide displacement [3,4]. In the prediction of related time series, the traditional idea is to carry the trend of past landslide displacements over time forward into the future, such as the moving and weighted moving average methods, exponential smoothing method, etc. In recent years, with the popularization of machine learning algorithms, various model methods have been applied in landslide prediction [5,6,7], such as ARIMA (Autoregressive Integrated Moving Average model), SVM (Support Vector Machine) [8] using the SVR algorithm for regression [9], RNN (Recurrent Neural Network), etc. In order to facilitate sequence modeling and have the ability of long-term memory, LSTM (long short-term memory) can be selected [10]. LSTM is a special RNN that largely alleviates the problems of vanishing and exploding gradients in deep learning model training. The LSTM itself has a relatively complex model structure and is suitable for the task of learning time series, keeping inputs for a long period of time and being able to handle time series better [11]. In 2022, Wang Y. et al. used LSTM, PSO-LSSVM and PSO-SVM models to compare the predictions of the Baijiabao landslide and concluded that the LSTM model had the highest prediction accuracy [12]. Therefore, this paper predicts based on the LSTM model with improvements in preprocessing to improve the prediction accuracy further.

In addition to the model’s method, the preprocessing method of the landslide monitoring data also affects the accuracy of the prediction results. When collecting monitoring point data in real-time, instrument failure or external influences may occur, resulting in some abnormal data. Therefore, the quality of the data interpolation method directly affects the accuracy of data collection. In recent years, there have been many interpolation algorithms used in related preprocessing, such as the Lagrange interpolation method, the Newton interpolation method with a mean difference, the linear interpolation method, the Hermitian interpolation method, etc. The cubic spline is effective in solving the image smoothness problem because of its good convergence and smoothness.

The traditional direct interpolation method is to bring the overall data into the interpolation algorithm so as to fit a smooth and complete time-displacement curve, and the interpolation value corresponding to any time can be obtained in the curve. However, due to the different effects of different interpolation algorithms, in order to ensure smoothness and accuracy, cubic spline interpolation is used for preprocessing, and then the ARIMA, univariate and multivariate LSTM models are used to predict and compare the effects and obtain a more suitable prediction model [6].

## 2. Materials and Methods

The Baijiabao landslide, the object of this interpolation algorithm research, is located in the Xiangjiadian Village, Guizhou Town, Zigui County, Hubei Province, China, on the right bank of the Xiangxi River. As shown in Figure 1, the landform around Baijiabao is a concave slope in the low mountain area; the front edge reaches the Xiangxi River, and the landslide shear outlet is located at an elevation of 125–135 m. The trailing edge of the landslide is bounded by bedrock, with an elevation of 265 m. The left side of the landslide is bounded by the lower bedrock of the ridge, and the right side is bounded by the ridge. The front edge is 500 m wide, the trailing edge is 300 m wide (both 400 m wide and about 550 m long), and the landslide area is 220,000 square meters. The plane shape of the landslide is short and tongue-like. The front edge of the deep sliding body is 20–30 m thick, the middle part is 47 m thick and the trailing edge is 10–40 m thick. The average thickness of the sliding body is 45 m, and the volume of the sliding body is 9.9 million cubic meters. The front edge of the shallow sliding body is 10 to 20 m thick, the middle part is 35 m thick and the trailing edge is 10 to 40 m thick. The average thickness of the sliding body is 30 m, and the volume of the sliding body is 6.6 million cubic meters. The climate here is a subtropical monsoon climate, with obvious canyon climate characteristics, abundant precipitation, four distinct seasons and an average annual rainfall of 1000–1400 mm. Most of the rainfall is concentrated in the summer, which is one of the areas with high incidences of rainstorms. The Three Gorges Canyon is deep, with strong winds and fixed wind direction, which is an east–west direction along the gorge. According to the exploration data, the soil material composition of the landslide here is composed of silty clay mixed with crushed rock soil, and the soil is loose and disordered [13,14]. As shown in Figure 2, according to the cross-section of the landslide body, the main components of the sliding bed material are composed of feldspar quartz sandstone and mudstone of the Jurassic Xiabadong Formation. The inclination of the rock formation is 260–285°, and the dip angle is 30–40°. Most of the front edge of the landslide is submerged in river water. When the water level of the reservoir rises to 175 m, part of the sliding mass is completely submerged in the water. The vegetation coverage rate of the mountain is good, mainly composed of subtropical broad-leaved forest and some mountain shrubs and grasses, but there are some exposed soil rocks in the upper part of the mountain, dozens of households living in the lower part and hundreds of acres of cultivated farmland [15]. There is also a Zixing highway passing through it. If a landslide occurs here, it will cause huge losses.

In order to ensure the safety of villagers’ lives and property on the Baijiabao landslide, researchers began to use GPS to monitor the surface displacement in October 2006. A total of four monitoring points have been constructed, namely ZG323, ZG324, ZG325 and ZG326. Among them, ZG324 and ZG325 are located on the main landslide surface, and ZG323 and ZG326 are located in the middle and lower part of the landslide near the Zixing highway. The displacement direction of these four monitoring points is consistent with the slope aspect; each monitoring point is monitored irregularly once a month and twice a month from June to September in the summer. Each instrument can be accurate to 10 decimal places [16].

From November 2006 to December 2012, the four displacement monitoring points (ZG323, ZG324, ZG325 and ZG326) set up in Baijiabao obtained 90 periods of data. The obtained data from the ZG323 monitoring point close to the highway was taken as the research object. According to the qualitative analysis of the Baijiabao landslide, it is believed that the rapid decline of the reservoir water level is the main influencing factor for the deformation of the Baijiabao landslide, and the rainfall in the same period accelerates the deformation of the landslide [17]. Therefore, we decided to consider rainfall and reservoir level factors in the multivariate LSTM predictions. The data for detecting landslide displacements is approximately once a month, while rainfall and reservoir water level data are detected and recorded daily so that the corresponding rainfall and reservoir levels can be found after landslide interpolation based on the available data. Since the displacement data of Baijiabao is a monthly cumulative displacement, the interval between the values is relatively large. First, the data is converted into a daily average displacement, and then the monthly rainfall and reservoir water level data of the corresponding region and date are obtained. All the data are shown in Figure 3 [18].

### 2.1. Cubic Spline Interpolation 

This study uses a cubic spline interpolation algorithm. Suppose that in the landslide time-displacement curve, the time interval [*a*, *b*] is divided into *n* intervals on average, and a smooth curve is constructed through these *n* points to fit the staged interpolation function.
(1)$$a=x_{0}<x_{1}<\ldots <x_{n}=b$$

For analyzing the displacement-time data of monitoring points, each time and the corresponding monitoring value can be used as a coordinate point, where *x* is the time on the horizontal axis, *y* is the monitoring value on the vertical axis and all monitoring data can be represented by (*x*,*y*). After the *n* monitoring points are segmented, (*x*_0_,*y*_0_), (*x*_1_,*y*_1_), …, (*x_n_*,*y_n_*), a polynomial interpolation equation is used to fit each sub-interval. Due to the low curvature of the first-order polynomial and the second-order polynomial, the third-order polynomial is used in [*x_i_*_−1_,*x_i_*], with continuous second-order derivatives [19]:(2)$$S_{n}(x)=\{s_{i}(x)|s_{i}(x)=a_{i}x^{3}+b_{i}x^{2}+c_{i}x+d_{i},\;x\in [x_{i-1},\;x_{i}],\;i=1,\;2,\;\ldots ,\;n\}$$

Since the cubic spline interpolation algorithm not only needs to be segmented but also needs to be smooth on the connected nodes at the district level, we can obtain
(3)$$\begin{array}{c} s_{i\;}\left(x_{i}\right)=s_{i+1}\left(x_{i}\right),\\ {s^{\prime }}_{i\;}\left(x_{i}\right)={s^{\prime }}_{i+1}\left(x_{i}\right),\\ {s^{\prime \prime }}_{i}\;\left(x_{i}\right)={s^{\prime \prime }}_{i+1}\left(x_{i}\right) \end{array}$$

With Formula (3), we set the limit conditions of segment endpoints, determine the natural, fixed and non-node boundaries and use the undetermined coefficient method to calculate the unknown values in Formula (2). Joining the curves of all the subintervals together will result in a perfectly smooth interpolated fit, with some improvement in accuracy.

After the displacement data of Baijiabao is subjected to cubic spline interpolation, some results are shown in Table 1 [20]. We will explore the validity of the interpolation in Section 4.

### 2.2. Autoregressive Integrated Moving Average Model (ARIMA)

The ARIMA model is one of the main models for time series forecast analysis. In ARIMA (*p*, *d*, *q*), *p* is the number of autoregressive terms, *q* is the number of moving average terms, and *d* is the number of differences made to make it a stationary sequence, also called the order. Even though the word “difference” does not appear in the English name of ARIMA, it is a crucial step in the main steps of using the ARIMA model [21]. The ARIMA model formula is shown in (4).
(4)$$x_{t}=\varphi_{0}+\varphi_{1}x_{t-1}+\varphi_{2}x_{t-2}+\ldots +\varphi_{p}x_{t-p}+a_{t}+\theta_{1}a_{t-1}+\theta_{2}a_{t-2}+\ldots +\theta_{p}a_{t-p}$$
where $a_{t}$ is the residual series, *p* denotes the number of autoregressive terms, *q* denotes the number of moving average terms, $\varphi_{0},\varphi_{1},\varphi_{2},\ldots ,\varphi_{p}$ denotes the autoregressive coefficients to be estimated, and $\theta_{1},\theta_{2},\ldots ,\theta_{p}$ denotes the moving average coefficients to be estimated. When *q* or *p* = 0, it can be transformed into an AR or a MA model. Since the ARMA model can only deal with process-smooth time series, the original series needs to be transformed into a smooth series by differencing it and then analyzed with the ARMA model to analyze non-smooth time series [22].

The difference is a mathematical tool that generally refers to the subtraction of the previous data from the following data. When the distances between them are equal, we perform this operation on each data and subtract the previous data from the following data, which is the “first order difference”. If all the above operations are repeated again, this is the “second order difference”.

As shown in Table 2, the equality of distance means that the previous data can be subtracted from the next data in rows 2, 3, 4, 5 and 6 or that the data in rows 2, 4 and 6 can be used for first-order difference operations, but the data in rows 1, 2 and 6 cannot be used because the distances are not equal [23].

The main steps of building an ARIMA model are as follows: the acquisition of the time series comes first before the preprocessing of the time series begins. The stationarity and white noise tests are methods for judging whether the data can be analyzed and predicted by the ARMA model [24]. The preprocessing of time series includes these two aspects of the test. Among them, testing data stationarity is a key step in time series analysis and methods, such as flow charts or ADF, which can be used to test the stationarity of the time series. In this study, we first draw a picture and substitute the data for rough judgment, and then use the ADF method to test. If the test results do not meet the requirements, the data is not stationary and needs to be differentially processed, and then the ADF test is used until the data is stationary. In theory, the extraction of non-stationary and deterministic information from time series information can be differenced three or more times, but in practice, this is not the case. Every time the difference method is used, a small part of the data will have truncation errors. Therefore, the difference method needs to be stopped in time and should not be overused. Therefore, the order of the difference is usually no more than two. Finally, it identifies one of the existing models that fit the obtained time series, and the model is ordered. The BIC criterion method can be used to calculate the order of the model. This step is to determine the values of *p* and *q* in the model. It can also be determined by drawing autocorrelation and partial autocorrelation diagrams. The standards are shown in Table 3. Both methods were used in this study. Fifth, correlation moment estimation and maximum likelihood estimation are common methods for estimating the parameters of the model. Sixth, the difference between the training results of the test model and the original is the verification of the model.

### 2.3. Long Short-Term Memory (LSTM)

LSTM can be simply understood as an RNN with more complex neurons. When processing time series with long intervals and delays, LSTM is usually better than RNN [25]. Compared with RNN neurons with simple structures, neurons of LSTM are much more complex. Each neuron accepts a cell state in addition to the sample input at the current moment and the output at the last moment. The neuron structure of LSTM is shown below in Figure 4: X represents the scaled information, + represents the added information, *σ* represents the Sigmoid layer, tanh (the hyperbolic tangent) represents the tanh layer, *h_t_*_−1_ represents the output of the previous LSTM unit, *C_t_*_−1_ represents the memory of the previous LSTM unit, *x_t_* represents the input, *C_t_* represents the latest memory and *h_t_* represents the output [26].

There are three gates in the LSTM neuron, namely the forget gate (*f_t_*), the input gate (*i_t_*), the output gate (*o_t_*) and the subscript t represents the time.

Forgetting gate (*f_t_*): *x_t_* and *h_t_*_−1_ are input and output, respectively. A value between 0 and 1 is used to determine how much to retain the cell state (*C_t_*_−1_) at the last moment, where 1 means fully reserved and 0 means fully abandoned. The specific formula is shown in (5). Taking this experiment as an example, when the LSTM model fits and predicts the displacement data in 2008, the output of the displacement data in 2007 through the forget gate is from 0.9 to 1, and the LSTM model will receive this data. However, if the displacement data in 2012 is predicted, then the output of the displacement data in 2007 through the forget gate should be infinitely close to zero. At this time, the LSTM model will discard the data, or it has been discarded as early as when it passed the forget gate.
(5)$$f_{t}=\sigma (W_{f}x_{t}+U_{f}h_{t-1})$$

Input gate (*i_t_*): First, use the Sigmoid layer, that is, the input gate, to decide which information needs to be updated in time, and then create a vector in the tanh layer, which includes how much information about the input value (*x_t_*) of the network can be stored in the cell state (*C_t_*) at the moment. Then, combine these two parts to update the information of the cell state (*C_t_*) at the moment. The specific formula is shown in (6). Taking this experiment as an example, when predicting the displacement data in December 2008, the input at this moment includes the data from June to November 2008. At this time, it is necessary to judge whether every data can be retained and then updated to the current cell state according to the data that can be retained.
(6)$$\begin{array}{c} i_{t}=\sigma (W_{i}x_{t}+U_{i}h_{t-1})\\ {\tilde{C}}_{t}=\tanh(W_{c}x_{t}+U_{c}h_{t-1})\\ C_{t}=f_{t}*C_{t-1}+i_{t}*{\tilde{C}}_{t} \end{array}$$

Output gate (*o_t_*): The Sigmoid layer, also known as the output gate, is used to determine how much cell state (*C_t_*) at the current time can be retained in the current output (*h_t_*), and then tanh is used to process the cell state. The product of two parts of information is the information to be output. The specific formula is shown in (7).
(7)$$\begin{array}{c} {ο}_{t}=\sigma (W_{o}x_{t}+U_{o}h_{t-1})\\ y_{t}=h_{t}={ο}_{t}*\tanh(C_{t}) \end{array}$$

Overall, the forget gate controls how much historical information has an impact on the present and the future, that is, how much can continue to be retained in long-state information. The input gate controls how much input information can be added to the long-state information, and the output gate controls how much of the aggregated information can be used as the current output [27].

### 2.4. Error Analysis Index

In this paper, the mean square error (MSE), root mean square error (RMSE) and mean absolute error (MAE) are used to evaluate the error of the final prediction result.

Mean Squared Error: the mean of the sum of the squares of the corresponding point errors of the predicted and true values, as shown in Formula (8).

Root mean square error: the square root of the ratio of the squared sum of the deviation between the predicted and true values to the number of samples (*n*), and measures the deviation between the observed and true values, as shown in Formula (9).

Mean absolute error: the true value minus the predicted value, which is then squared and averaged for the average of the absolute errors, as shown in Formula (10).
(8)$$\mathrm{MSE}=\frac{\sum {\left(x-x_{i}\right)}^{2}}{n}$$
(9)$$\mathrm{RMSE}=\sqrt{\frac{\sum {\left(x-x_{i}\right)}^{2}}{n}}\;$$
(10)$$\mathrm{MAE}=\frac{1}{n}\sum_{i=1}^{n}\left|x-xi\right|$$

In the formulas, *x* and *x_i_* are the actual and predicted values, respectively, and *n* is the number of samples.

## 3. Results

This chapter introduces the experimental procedure in detail and analyzes the results. The process is shown in Figure 5. Firstly, the displacement data of Baijiabao is preprocessed by cubic spline, and then the ARIMA, univariate and multivariate LSTM models are used for displacement predictions. Finally, the error analysis of the prediction results of the three models is carried out.

### 3.1. ARIMA Prediction

The original displacement data and the corresponding first-order and second-order difference plots are shown in Figure 6. It can be seen from the figure that the fluctuation of the original data is large and unstable, and differential processing is required. Most of the data in the first-order difference fluctuate around zero, and only a few greatly fluctuate and feel stable. The second-order difference data fluctuates around zero and are judged to be stable data.

In order to assess the data more accurately, several orders of difference and the ADF unit root test methods should be used. The results are shown in Table 4.

The ADF result includes five parts of data. The first part is the result of the ADF test, abbreviated as the t-value, which represents the t statistic; The second part is referred to as the p-value, which represents the probability value corresponding to the t statistic; The third part represents the delay; The fourth part represents the number of tests; The fifth part is viewed together with the first and is the value of the critical ADF test under the 99, 95 and 90% confidence intervals. The criterion for stationarity is that the t-value is less than the critical ADF test value under the 99% confidence interval, and the *p*-value is less than 0.05, preferably approaching zero [28].

According to the ADF test standard, the calculated first-order difference ADF results are consistent. Therefore, the first-order difference achieves the purpose of stationarity, and the difference value “d” of the model takes a value of one.

There are many ways to determine the *p* and *q* values in the ARIMA model. This paper uses the following two methods to find more suitable *p* and *q* values.

One is to observe the autocorrelation (ACF) plot and the partial autocorrelation (PACF) plot. The upper part of Figure 7 is the ACF map, and the lower part is the PACF map. It can be observed that they are all trailing features when the abscissa is greater than two, the ordinate always has a non-zero value, and it will not always be equal to zero after the abscissa is greater than a certain constant. Therefore, the *p* and *q* values of the model are determined to be two.

The second is the BIC criterion. The judgment criterion is that the smaller the BIC value, the better. The values of *p* and *q* are limited to 0 to 5, respectively, according to the values determined in the previous ACF and PACF diagrams. The calculated BIC heat map is shown in Figure 8. The smaller the value, the darker the color and the more consistent it is. Then, the output of the *p* and *q* values determined by BIC is also 2 and are compared with the determination results of AIC, as shown in Figure 9. The *p* and *q* values are both taken as 2 when combining the two methods.

To sum up, the ARIMA (p, d and q) model conforming to the Baijiabao displacement data is ARIMA (2, 1 and 2). The displacement data is divided into a training and prediction set. Here, the last eight periods are predicted and used as the prediction set, and the previous historical data is the training set. The results of the fitting predictions model are shown in Figure 10. The RMSE of the fitting part was 0.336, and that of the predicted part was 3.37. The error of the prediction part is ten times that of the fitting part. It can be seen that the fitting effect of the ARIMA model is very good, and the value of the prediction part is too large. Because of the limitations of the ARIMA model, the inflection point cannot be well predicted.

### 3.2. Univariate LSTM Prediction

In this experiment, the displacement data is divided into a training and prediction set according to the ratio of 9:1. To determine the appropriate training number, the training epoch is taken as 50 and then the training set is continuously trained, and the value of the corresponding loss function is calculated after each iteration of training. Finally, the value of the smaller loss function and the relatively small number of training epochs is taken as the number of training epochs. The loss graph is shown in Figure 11. After testing, it is determined that the number of training rounds is 26. After debugging, the following parameters were derived: n_epochs = 26, batch_size = 64 and n_timestamp = 6.

The prediction set was input into the trained model, and the last eight periods of displacement data were predicted. The fitted predictions are shown in Figure 12. It can be observed that as the number of samples increases, the fitting effect is better and better, and the prediction part is very close to the real data. Table 5 shows the corresponding errors of different sample stages. After the calculation of different sample number intervals, it is finally determined to use 27 as the sample interval for error calculation, which further verifies the previous observation that with the increase of the number of samples, the fitting effect is better and better.

### 3.3. Multivariate LSTM Prediction

In Figure 3, it can be observed that the monthly rainfall from May to September is relatively large, and the reservoir water level has a small rise around October of every year. In conjunction with the displacement, it is observed that the water level of the reservoir has a small increase after a large amount of rainfall, and each rising interval of the displacement is almost when the monthly rainfall is relatively large. It can be seen that there is a certain relationship between monthly rainfall, reservoir water level and displacement. Therefore, it was decided to add monthly rainfall and reservoir water level data to the univariate LSTM model to form a multivariate LSTM model [29].

After adding the corresponding monthly rainfall and reservoir water level data on the basis of the univariate LSTM model, the data are divided into training, test and prediction sets according to the ratio of 7:2:1. To determine the appropriate training number, the training epoch is taken as 50, then, the training set is continuously trained, and the value of the corresponding loss function is calculated after each iteration of training. Finally, the value of the smaller loss function and the relatively small number of training epochs is taken as the number of training epochs [30]. The loss graph is shown in Figure 13. After testing, the number of training rounds was determined to be 20. After debugging, the following parameters were derived: epochs = 20, batch_size = 72 and n_timestamp = 5.

The test and prediction sets were put into the trained model to predict the last eight periods of displacement data, and the training test and prediction results are shown in Figure 14. It can be observed that both the test and the prediction parts are very close to the real data.

## 4. Discussion

The prediction results of the ARIMA, univariate and multivariate LSTM models are compared and analyzed together with the original data. The prediction results are shown in Figure 15, the percentage error is shown in Figure 16, and the error evaluation index results are shown in Table 6. To demonstrate the feasibility of cubic splines for this data, we added predictions using a multivariate LSTM model for the original data without interpolation. As shown in Table 6, the mean squared error (MSE) was 1.449, the root mean squared error (RMSE) was 1.204 and the mean absolute error (MAE) was 1.044, which is greater than the prediction error using the multivariate LSTM model for the interpolated data.

It can be seen from Figure 15 and Figure 16 and Table 6 that, among the three prediction models, the displacement prediction results of the ARIMA model are the most different from the original data, and the prediction effect is the worst. The fundamental reason is that the ARIMA model itself has limitations. The fitting effect is good, but the prediction requirements for the stationarity of the data are too high, and the inflection point cannot be well predicted. Therefore, the ARIMA model is not a good landslide displacement prediction model.

The displacement prediction effect of the univariate LSTM model is better than that of the ARIMA model because the model fully mines the variation law of historical displacement data and selects historical data that has an impact on the current prediction results for prediction. Therefore, compared with the ARIMA model, this model has a better prediction effect and is a more suitable model for landslide displacement prediction.

Among the three models, the multivariate LSTM model has the best prediction effect, and the prediction results are comparatively more accurate. It not only fully mines the variation law of historical displacement data but also mines the relationship between historical monthly rainfall, reservoir water level and displacement data [31].

The mean square error of the prediction result is 0.64223 mm, the root mean square error is 0.801 mm, and the mean absolute error is 0.505 mm, which is smaller than 10.241, 3.200, and 3.121 mm of the ARIMA model and 1.126, 1.061, 0.704 mm of the univariate LSTM model, respectively. This shows that the LSTM model has certain advantages in data mining after the introduction of influencing factors, and the multivariate LSTM model is a more suitable model for landslide displacement prediction [32].

## 5. Conclusions

In order to better prevent landslide disasters, the study of landslide displacement is crucial. The research object selected in this paper is the displacement data of the Baijiabao landslide. First, the cubic spline preprocessing method is used to interpolate the missing data, and then the ARIMA and LSTM models are used to predict the displacement data.

This paper first compares the univariate LSTM model with the univariate ARIMA model. The former model has comparatively smaller mean square, root mean square and mean absolute error values in its data. So, compared with the ARIMA model, the univariate LSTM model is better. This may be because the ARIMA model has certain limitations and cannot predict the inflection point of the displacement curve well, and the error sharply increases when the trend changes, resulting in a large deviation between the prediction results and the actual situation. Then, the rainfall and reservoir water level data are introduced into the input of the LSTM model, thereby improving the prediction accuracy of univariate LSTM models. The experimental results show that compared with the univariate LSTM model, the mean square error, root mean square error and mean absolute error data of the multivariate LSTM model are comparatively smaller. Therefore, it can be concluded that the multivariate LSTM model can better predict the displacement of the Baijiabao landslide than the univariate LSTM model, and the predicted value of the model is closer to the true value. The features of rainfall and reservoir water level data are added to the multivariate LSTM model, indicating that these two data affect the displacement data to a certain extent. The multivariate LSTM model does not simply limit the impact of historical displacement data on the whole during prediction but integrates historical rainfall and reservoir water level data to make the displacement weight corresponding to each variable more accurate, so the prediction effect is comparatively better.

The Baijiabao landslide displacement data are stepped and sensitive to rainfall and reservoir levels. In this paper, we used a cubic spline interpolation method to preprocess the stepped Baijiabao landslide displacement data and a multivariate LSTM model to predict the Baijiabao landslide displacement. Compared with the interpolated ARIMA, the univariate and multivariate LSTM models without interpolation yielded lower prediction errors. The combined method provides a new idea for predicting step-like types of landslides.

## Figures and Tables

**Figure 1 ijerph-20-01167-f001:**
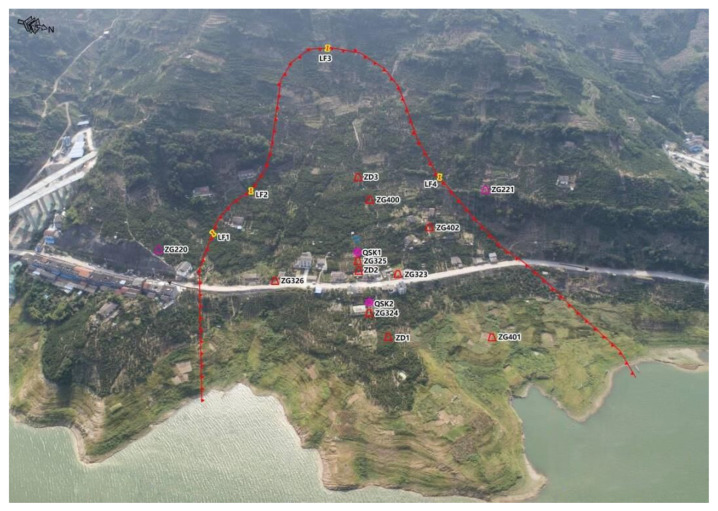
Full view of Baijiabao.

**Figure 2 ijerph-20-01167-f002:**
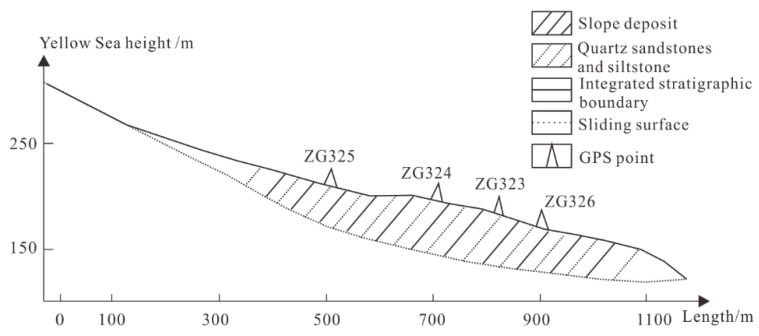
Sectional map of the Baijiabao landslide, showing the distribution of soil quality and the location of GPS detection.

**Figure 3 ijerph-20-01167-f003:**
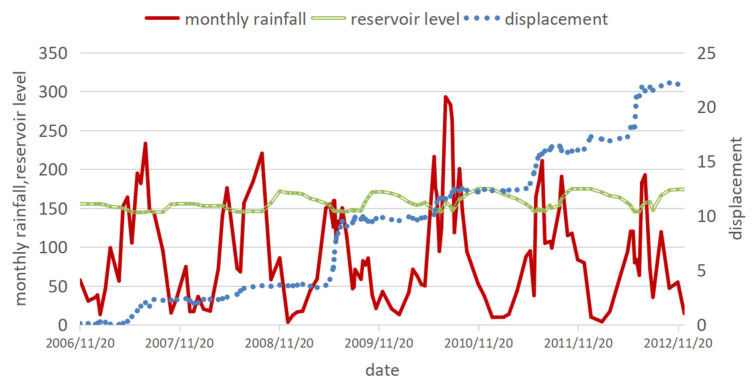
Data map of rainfall, reservoir water level and displacement.

**Figure 4 ijerph-20-01167-f004:**
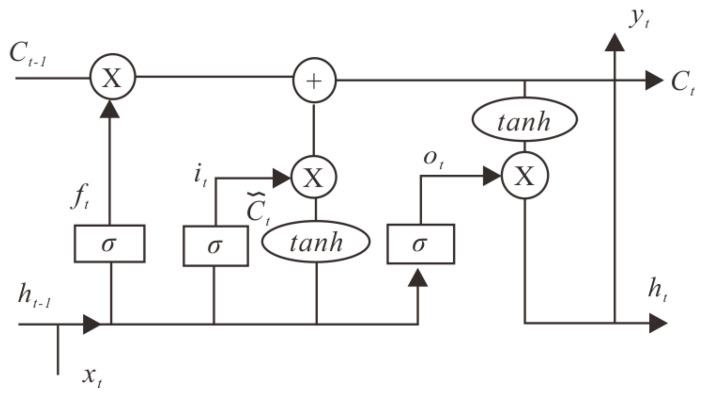
Structure chart of LSTM.

**Figure 5 ijerph-20-01167-f005:**
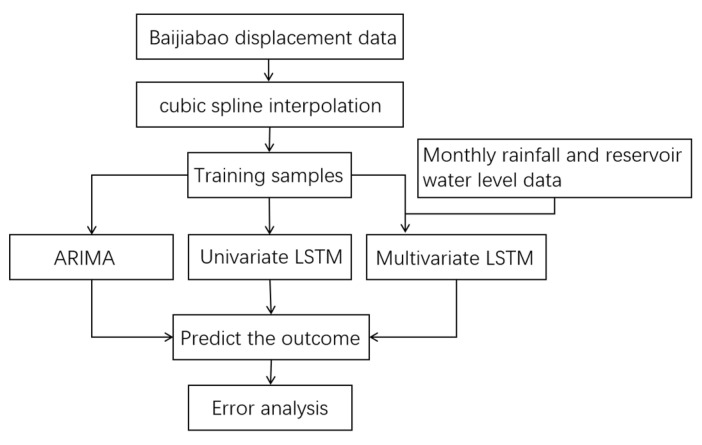
The research process of Baijiabao’s displacement prediction.

**Figure 6 ijerph-20-01167-f006:**
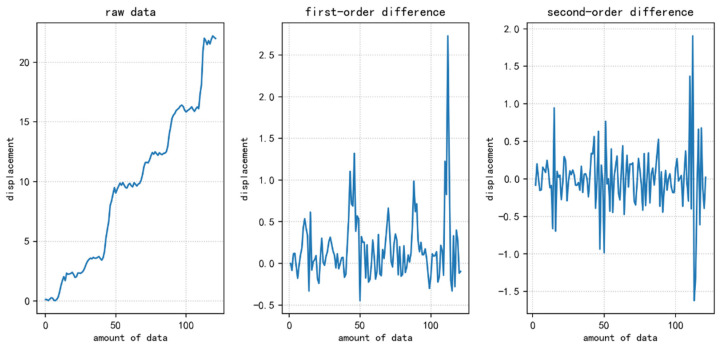
The difference diagrams are the original data, first-order and second-order difference data from left to right.

**Figure 7 ijerph-20-01167-f007:**
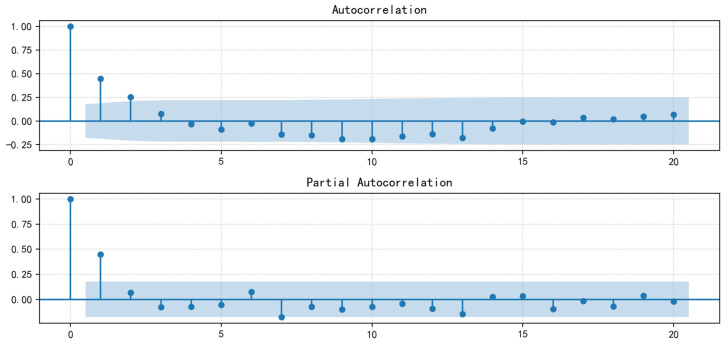
ACF and PACF diagrams.

**Figure 8 ijerph-20-01167-f008:**
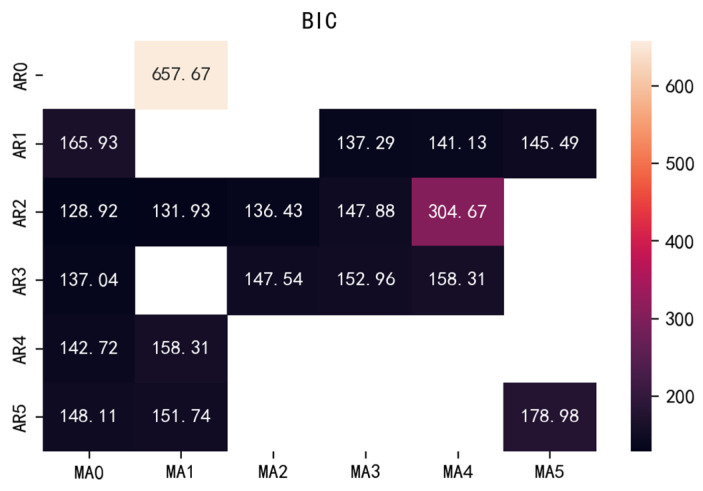
BIC thermodynamic diagram.

**Figure 9 ijerph-20-01167-f009:**
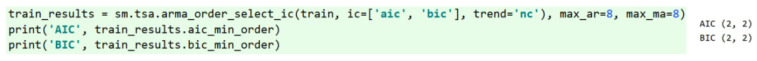
BIC code and results.

**Figure 10 ijerph-20-01167-f010:**
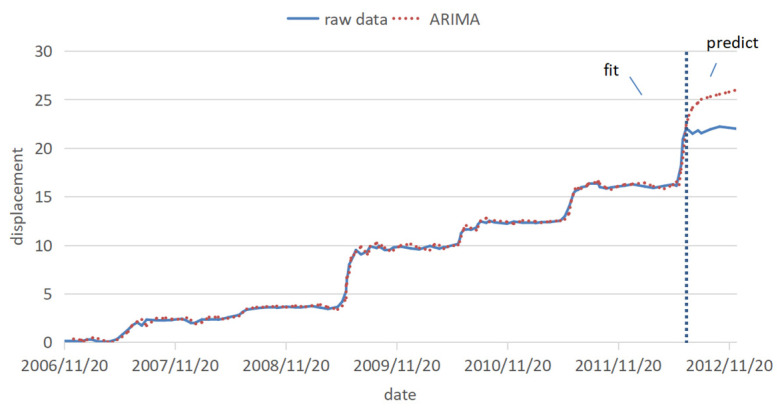
Fitting prediction results of ARIMA model.

**Figure 11 ijerph-20-01167-f011:**
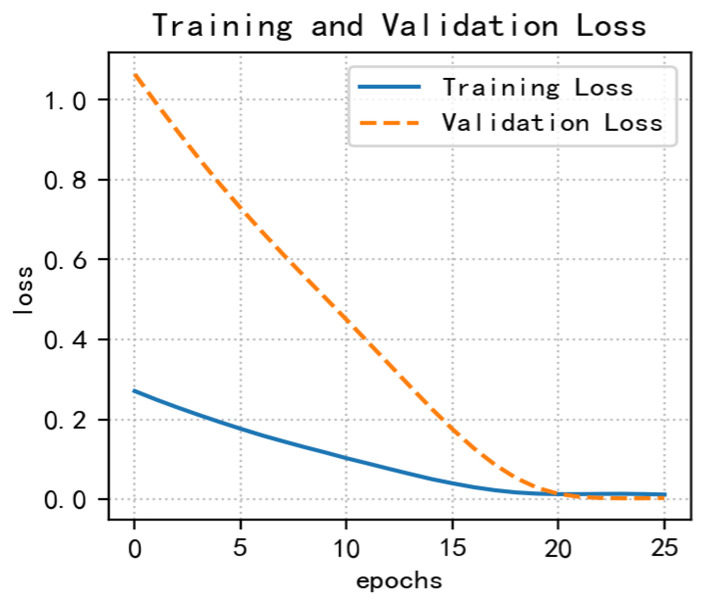
The research process of Baijiabao’s displacement prediction.

**Figure 12 ijerph-20-01167-f012:**
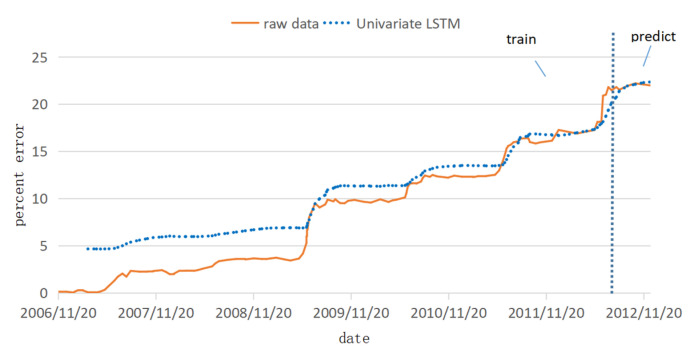
The fitting and prediction result of the univariate LSTM model.

**Figure 13 ijerph-20-01167-f013:**
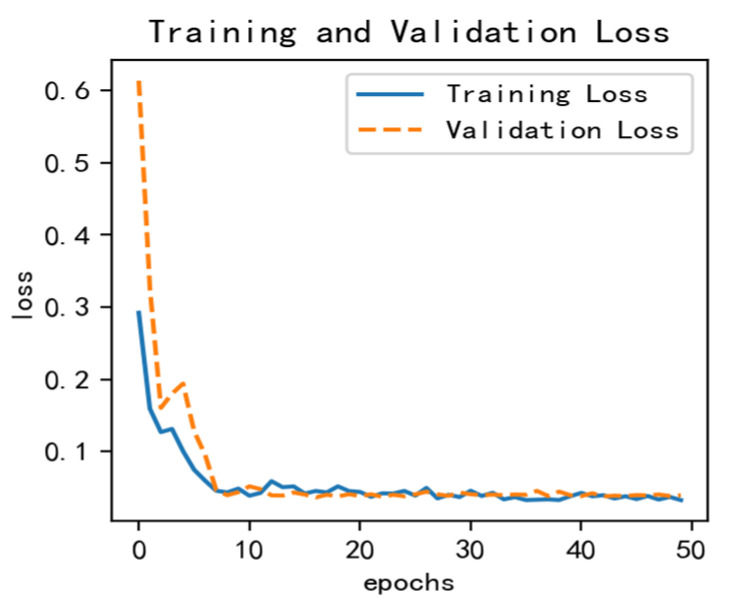
Multivariate LSTM model loss diagram.

**Figure 14 ijerph-20-01167-f014:**
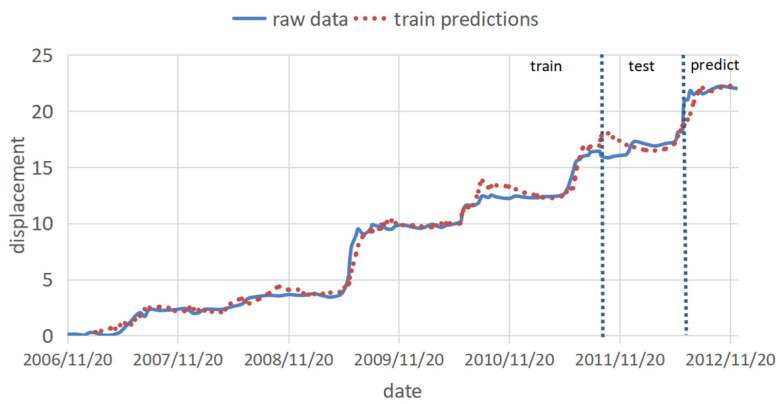
Multivariate LSTM model training prediction diagram.

**Figure 15 ijerph-20-01167-f015:**
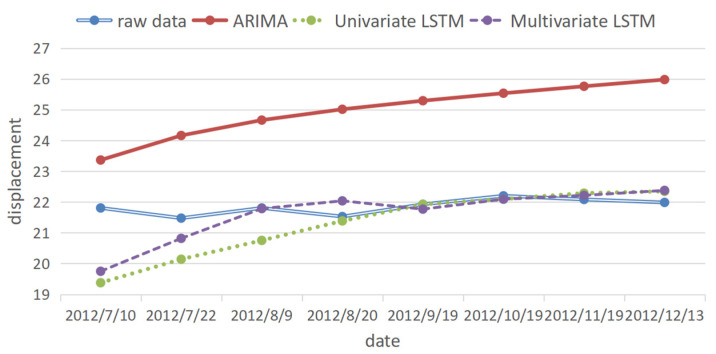
Prediction results of the three models.

**Figure 16 ijerph-20-01167-f016:**
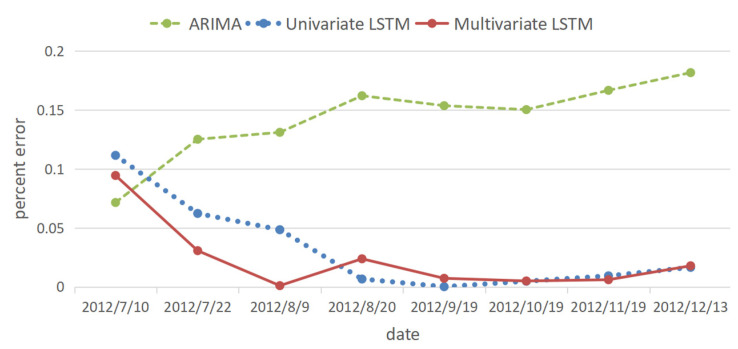
Percentage error plot of the three models.

**Table 1 ijerph-20-01167-t001:** Interpolated displacement data comparison table.

Raw Data of ZG323 (mm)	Cubic Spline Interpolation Data of ZG323 (mm)	Coefficient Matrix of Cubic Spline
…	…	$\left[\begin{array}{cccc} \cdots & \cdots & \cdots & \cdots \\ 0.00011 & 0.00193 & 0.11262 & 1.11803\;\\ -0.00013 & 0.01207 & 0.54638 & 9.70824\\ -0.00008 & -0.00245 & 0.89264 & 38.74597\\ \cdots & \cdots & \cdots & \cdots \end{array}\right]$
0.037	0.037	
	0.137	
0.324	0.324	
	0.756	
1.292	1.292	
	1.711	
2.042	2.042	
…	…	

**Table 2 ijerph-20-01167-t002:** Example of the first-order difference method.

Serial Number	Raw Data	Data after First-Order Difference
1	11	null
2	12	1
3	13	1
4	14	1
5	15	1
6	16	1

**Table 3 ijerph-20-01167-t003:** Model order table.

Model	ACF	PACF
AR (*p*)	Trailing	Truncation
MA (*q*)	Truncation	Trailing
ARMA (*p*, *q*)	Trailing	Trailing

**Table 4 ijerph-20-01167-t004:** The results of the ADF unit root test.

	Raw Data ADF	First-Order Difference Data ADF
t	0.065	−6.730
p	0.964	3.311 × 10^−9^
Delays	1	0
Number of tests	120	121
1%	−3.486	−3.486
5%	−2.886	−2.886
10%	−2.560	−2.580

**Table 5 ijerph-20-01167-t005:** Corresponding errors of different sample stages.

Sample	MSE (mm)	RMSE (mm)	MEA (mm)
1–27	12.962	3.600	3.563
28–54	4.624	2.150	1.824
55–81	1.296	1.138	1.046
82–108	0.797	0.893	0.642

**Table 6 ijerph-20-01167-t006:** Corresponding errors of the three models.

	ARIMA	Univariate LSTM	Multivariate LSTM (Without Interpolation)	Multivariate LSTM
MSE (mm)	10.241	1.126	1.449	0.642
RMSE (mm)	3.200	1.061	1.204	0.801
MAE (mm)	3.121	0.704	1.044	0.505

## Data Availability

The data presented in this study are available on request from the corresponding author.

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
