# Peer review of "Landslide Displacement Prediction Based on Multivariate LSTM Model"

_ijerph, 2023, doi:10.3390/ijerph20021167_

Round 1

Reviewer 1 Report

See comments and suggestions in the attached file.

Reviewer 2 Report

The paper deals with the prediction of landslide displacement using multivariate LSTM.
It has merit and deserves publication on the Journal, with a minor, mandatory review, which follows:

1) In the multivariate LSTM, the monthly rainfall and reservoir water level are considered. It is suggested to detail how these covariates are incorporated into the model.

2) Line 182, citation to the reference is missing.

3) Figure 4, the "triple spline pretreatment" should be "cubic spline interpolation".

Reviewer 3 Report

1. In part 1, the author does not clearly introduce the advantages of  preprocessing method and deep learning methods used in paper. It is suggested that a brief description of the author's contribution should be included in this part.

2. In 2.1, when the cubic spline interpolation method is used for data preprocessing, all the data are only piecewise interpolated as training data, but the data set is not verified to verify whether the method is effective for this kind of landslide displacement data. It is suggested to add verification data sets to verify the correctness and effectiveness of the piecewise cubic spline interpolation method.

3. In 2.1, It is suggested that the author should not only list the interpolated data in Table 1, but also list the coefficients of the equation obtained by segments.

4. In 2. 2, it is also recommended that the author add a detailed description of the ARIMA coefficient.

5. The author uses multivariable LSTM network to predict landslide displacement, but the author firstly interpolates the landslide displacement, so the amount of displacement samples is more than that of rainfall and reservoir level data. It is suggested that the author describe in detail the construction process of the above data set, whether the same interpolation is carried out for environmental factors, or is the same amount of landslide displacement data selected as that of environmental factors?

6. It is suggested that the author can list the parameters and super parameters of the deep learning network in order to reproduce later.

7. How to determine the optimal parameters of the proposed model,  such as why 50 rounds of training are selected, and whether the optimization parameter selection algorithm is used?

8. The author only compares three classical prediction methods proposed by previous researchers, but there are many other prediction methods in Baijiabao area at present now. It is suggested that the author should add the comparison with other advanced methods to enhance the persuasiveness of the method proposed in this paper.

9. In this article, both the data preprocessing method and the displacement prediction method are more mature methods, so it appears that the author's innovation and contribution are not enough. It is suggested that the author should further refine the innovation of the article to make the article more frontier and innovative.

Round 2

Reviewer 3 Report

The author made detailed amendments according to the opinions, which greatly improved the prudence and reproducibility of the article. And some confused points also carried on the further description, which solved my doubt. Therefore, it is recommended to publish.